# Evaluation of a service introduced to support young people at risk of suicide using a cohort design, a mixed methods analysis and cost-benefit analysis

**Denny Meyer**[1]*, **Liza Hopkins**[2], **Michelle Kehoe**[2,3], **Richard Whitehead**[1,2], **Kathleen de Boer**[1], **Debra Osborne**[1], **Maja Nedeljkovic**[1]

**1** Centre for Mental Health and Brain Science, Swinburne University of Technology, Melbourne, Australia, **2** Alfred Mental and Addiction Health, Alfred Health, Melbourne, Australia, **3** School of Primary and Allied Health Care, Monash University, Melbourne, Australia

* dmeyer@swin.edu.au

**Data Availability Statement:** The data contain potentially identifying or sensitive patient information. Restrictions on the sharing of data

## Abstract

This evaluation sought to determine to what extent a suicide prevention service, designed for young people at severe risk of suicide, was successful in reducing suicide risk and distress and improving well-being outcomes. The 3-month service was co-designed at the height of the COVID-19 pandemic in Melbourne, Australia, with young people and carers who had lived experience of youth suicide. The evaluation involved a mixed methods analysis of data collected during the first 20 months of operation, from February 2022 to September 2023. Ninety one young people were admitted to the service during this period, of whom 28 agreed to participate in the evaluation. Primary data were collected by telephone or online, and secondary data for hospital service use were collected for 70 of the young people. Analysis found that there were significant reductions in suicide risk and psychological distress at discharge, associated with significant improvements in quality of life and reductions in the use of hospital services. Qualitative analysis supported these survey results, highlighting the peer and psycho-social support as particularly beneficial aspects of the service. However, the qualitative analysis also indicated that some individuals felt the service was too short and abruptly ended, with insufficient attention paid to the creation and maintenance of support services that could continue after discharge. A small sample size was a limitation of this study, particularly for the three and six month follow-up surveys. Also, the cost-benefit analysis included secondary routine hospital admission data covering only 12 months. Nevertheless, the overall findings are positive, suggesting that services of this nature can have real benefits for young people at serious risk of suicide. It is likely that an extension of the cost-benefit analysis, to allow for benefits beyond a one year period, would have suggested that the benefits outweigh the costs.

have therefore been imposed by the Alfred and Swinburne University of Technology Ethics Committee. However, limited data may be made available on request with requests sent to the Swinburne University of Technology Human Ethics Committee (resethics@swin.edu.au).

**Funding:** This work was supported by Suicide Prevention Australia Innovation Grant to DM [grant no. RFF/20/261]. The funders had no role in study design, data collection and analysis, decision to publish, or preparation of the manuscript.

**Competing interests:** The authors have declared that no competing interests exist.

## Introduction

Suicide is now the leading cause of death among Australians aged 15–25 years old. In 2022, 64 deaths by suicide represented 30.9% of all deaths in young people aged 15–17, and 304 deaths by suicide represented 32.4% of all deaths in those aged 18–24 years, up from 16.5% and 23.9% respectively in 2001 [1]. These statistics underscore the urgency of developing and evaluating services designed to prevent suicide in suicidal young people.

The success of the Victorian State Government-funded Hospital Outreach Post-suicidal Engagement (HOPE) initiative for adults has prompted the introduction of similar services for young people [2]. In particular, the Royal Commission into Victoria's Mental Health System recommended creation of a new HOPE service for children and young people who have self-harmed or are at risk of a suicide attempt, with the ultimate goal of reducing youth suicide [3]. A study of adolescents has shown that more than half of individuals with deliberate self-harm present significant suicide risk, with self-harm behaviour pervasive in adolescents that attempt suicide [4]. Previous suicide attempts have also been recognised as a significant predictor of death by suicide [4].

However, it has been shown that for 632 young people aged 15–24 who survived their first suicide attempt, engagement in psychological support led to a significant reduction in suicide attempts, with only 1.4% of the young people subsequently dying from suicide [5]. A meta analysis has also indicated that interventions for the prevention of youth suicide are effective when delivered in clinical, educational and community settings, resulting in reductions in self-harm and suicidal ideation [6]. Findings in this meta-analysis indicated that brief contact interventions in clinical settings reduces the risk of a repeat suicide attempt by 11% post-intervention and by 17% after last follow-up assessment. In addition, this study indicated that psychoeducation combined with screening in school settings reduced risk by 69% post assessment and 37% for follow-up assessments. However, an older study from five low- and middle-income countries does not confirm the effectiveness of brief educational intervention and follow-up contacts in reducing subsequent repetition of suicide attempts after discharge from emergency departments, with a variety of possible reasons given for the inconclusive findings and differences across sites [7].

The recent literature on suicide prevention in young people has also considered web-based and mobile suicide prevention interventions. There is pervasive use of the internet and mobile phones by young people and a common reluctance of suicidal young people to seek professional help for their mental health problems [8, 9]. However, the overall effectiveness of eHealth interventions for suicide prevention in young people is yet to be confirmed [10, 11]. The heterogeneity of outcome measures, the tendency to exclude participants with elevated suicide risk, and the lack of standardised classification systems for such interventions have made an evaluation of the effectiveness of these interventions more difficult [12].

Other important areas of research in regard to suicide prevention in young people relate to the use of peer support approaches in both face-to-face and online contexts [13, 14]. Peer support specialists, who have lived experience of mental health issues, are thought to promote hope by providing role models for recovery and by providing emotional support, decreasing loneliness and stigma, and facilitating improved relationships with others [15]. However, very little is known about the effectiveness of this approach with few relevant evaluations conducted. This has resulted in a call for the reporting of key service characteristics and components and the need for more formal service evaluations of interventions providing peer support for suicidal young people [13]. The present study is a response to this call.

The purpose of this study is to evaluate a new service to which children and young people under the age of 26 are referred after a suicide attempt, self-harm or persistent suicidal

ideation. This service is being delivered within an existing public child and youth mental health service in Victoria, Australia. It encompasses time-limited support (maximum 3 months engagement) for young people aged under 26 years with an acute suicidal risk (recent attempt or persistent ideation), provided by a multidisciplinary team including psychiatric, psychological, psychosocial and lived experience staff. The team provides intensive outreach wrap around support to the young person (and their families where appropriate), includes liaison with other supports such as school wellbeing services, and concludes with a warm handover to ongoing supports where needed.

As recommended by the Royal Commission into Victoria's Mental Health System, a co-design process was used to design this service, in order to acquire a comprehensive understanding of user needs and a greater diversity of ideas for meeting these needs [16]. As reported elsewhere, co-design processes often pose implementation challenges, and the same was true for this program [17–19].

The importance of co-design for services of this nature, and the challenges in implementing such co-designed services, make an early evaluation of this service of particular importance. This evaluation has been specifically designed to determine if there was a reduction in suicidal ideation and improvements in coping, self-efficacy, resilience and well-being for the young people after engaging with the service, and if this was sustained three to six months following discharge, resulting in reductions in the usage of hospital services. In addition, the objective was to obtain feedback from the young people themselves regarding their experience of the service, and to conduct a cost-benefit analysis for the service. It is hoped that the results of this evaluation can be used to make improvements to the service itself and to assist with the design of future similar services, including web-based and mobile suicide prevention interventions.

## Materials and methods

### Ethics statement

This study was carried out in strict accordance with the ethics approval obtained from the Alfred Health and Swinburne University of Technology human research ethics committee. All participants older than 18 years of age provided written consent. Parental/guardian consent was obtained for participants under the age of 17. Particular care was taken to preserve the anonymity of participants in view of the small sample size, with sub-group reporting only when essential (e.g. when different measures were used).

Note on language: In this paper we refer to young persons with lived or living experience of mental health issues. We acknowledge that other terms are used such as a person with LLE (lived and living experience) and service user or consumer or client in different countries and jurisdictions. Those who care for the young person are referred to as parents, families or carers but we acknowledge that other terms may be used for unpaid carers throughout the world.

All primary and secondary data collection was conducted in Melbourne, Australia, with the secondary data consisting of routinely collected hospital service data. A quantitative analysis including an economic evaluation was conducted using data from both these data sources, while a qualitative data analysis was conducted using only primary data. The STROBE cohort checklist and the SRQR checklist guided reporting of the quantitative and qualitative results respectively [20, 21].

### Primary data collection

The service started accepting referrals for young people from the middle of February 2022, with 91 young people accepted before 30th June 2023. However, the recruitment of the 28 young people for the collection of primary data for this evaluation study began on 5th April

2022, allowing time for the service to establish itself. Primary data collection ended at the end of September 2023.

The effectiveness of the service was ascertained using a variety of age appropriate standard measures at up to four time points–entry to service, discharge, three-month follow-up and six-month follow-up. Quantitative and qualitative data were collected in either telephone interviews or via online surveys that could be completed at a time and day that suited participants. The mode of data collection was determined by participant preferences. Further qualitative data were also collected from three telephone interviews.

The evaluation team has worked in the area of suicide prevention for many years. They have some lived experience of suicide and some experience of a similar adult service (blinded for review) and suicide screening (blinded for review). This experience has informed the survey questions included in this evaluation study and has also influenced the way in which this evaluation has been conducted, with the utmost care being taken to ensure that the safety of the young people included in the study has been protected. The involvement of parents or carers for the young people under the age of 17 included in the evaluation, and the choice of developmentally appropriate scales for this group, provide examples of this approach. The choice of telephone as the medium for the collection of the survey data for most of the data collection (for all but some of the follow-up surveys), was also intended to ensure that data collection could be conducted in an appropriate manner that was also COVID-19 sensitive.

## Recruitment for primary data collection

The participants in the primary data collection included 28 young people who received the service between April 2022 and September 2023. Apart from a few young people under the age of 12 or young people who were judged by the clinical treating team to be too ill, all the young people who came through the service during this period were invited to participate in the evaluation. Potential participants were provided with a flyer explaining the purpose of the evaluation when they entered the service. Within the next week they were telephoned or texted by the research team, and asked if they were interested in joining the evaluation study. Up to three attempts were made to contact participants in this way, at each of the four timepoints. All those recruited met the inclusion criteria, having sufficient cognitive ability to understand the evaluation and consenting process, with an ability to read and understand written and spoken English.

Potential participants aged 17 or above were provided with a link to an explanation of the project, a Participant Information Sheet and a consent form, all written in simple English. For participants aged between 12 and 16 years, the participant's parent/guardian was approached and asked to provide consent for their child to participate. When consent was given, the children themselves were also asked to provide assent at the time of the data collection.

By agreeing to participate, each young person was agreeing to provide primary data at the four data collection timepoints described previously. They were also agreeing to be contacted by the research team to invite participation in a satisfaction survey and a qualitative interview at the 3-month or 6-month post-discharge timepoints. Interviews were also conducted with some family members/carers of consenting participants and these results will be reported elsewhere.

## Service Evaluation Measures (Primary data)

A broad range of measures, partly informed by the prior evaluation of results for the similar adult service, were chosen in order to capture the varied and complex presentations expected. The constructs measured included suicidality, hope, quality of life, coping self-efficacy,

resilience and well-being. Advice from two psychologists within the service was sought regarding potential participant burden. Subsequent RedCap data indicated that it generally took 15–16 mins (with a range of 5 to 23 mins) for the survey to be completed online. However, it took up to 40 mins for the survey to be completed over the telephone.

These measures, described below, were collected at entry to the service, at discharge and at 3-months and 6-months follow-up using summated scales.

a. The Adult State Hope Scale (AHS), consisting of six items, each recorded on an eight-point Likert scale (1–8), with higher values indicating greater hope, with demonstrated internal reliability ($\alpha$ = 0.74–0.88). The AHS measures Snyder's cognitive model of hope, defined as a positive motivational state [22]. This scale was used only for young people aged 17 or older at service entry.

b. Snyder's Children Hope Scale (CHE), consisting of six items, each recorded on a six-point Likert scale (1–6), with higher values indicating greater hope [23]. This scale is used to measure hope in individuals aged 8 to 16 with test-retest reliability over a one-month interval of 0.71 to 0.73. This scale was used only for young people aged 16 or younger at service entry.

c. The Outcome Rating scale (ORS), consisting of four items, each recorded on a 0–100 continuous scale, with higher values indicative of better well-being [24]. This scale measures the client's perspective of change or improvement (or lack thereof) in relation to where they started. It has demonstrated excellent internal consistency ($\alpha$>0.9) and test-retest reliability [25]. This measure was used for all the young people.

d. The Coping Self-Efficacy scale (CSE), consisting of 26 items, all recorded on a 10-point Likert scale (1–10), with higher values associated with greater belief in an ability to perform behaviours important for adaptive coping [26]. This scale has demonstrated excellent reliability ($\alpha$ = 0.91). This measure was used for all the young people.

e. The EuroQol-5 Dimension (EQ-5D) quality of life scale, consisting of five items recorded on a 3-point Likert scale (1–3), with higher values indicating poorer quality of life [27]. This scale has been validated with adolescents and young adults with posttraumatic stress disorder [28].

f. The Patient Health Questionnaire (PHQ-A) for adolescents, consisting of nine items recorded on a 4-point Likert scale (0–3), with higher values indicating higher levels of depression. This scale was used only with young people aged 16 or younger at service entry. A categorical scoring system was used for this measure with scores less than 5 associated with an absence or minimal depressive disorder, 5–10 mild, 11–14 moderate, 15–19 moderately severe and 20–27 severe depression [29, 30].

g. The Patient Health Questionnaire (PHQ4) psychological distress scale, consisting of 4 items recorded on a 4-point Likert scale (1–4), with higher values indicating more distress. This scale was used only for young people over the age of 17 at service entry. The first two items are associated with anxiety and the last two items are associated with depression [31].

h. The Suicidal Ideation Attribute Scale (SIDAS) consisting of 5 items recorded on an eleven-point Likert scale (0–10), with higher values indicating more suicidal ideation (one item reversed). This scale was used only for young people over the age of 17 at service entry. The SIDAS is designed to screen individuals in the community for the presence of suicidal thoughts and to assess the severity of these thoughts [32]. It has high internal consistency ($\alpha$ = .89) and good test-retest reliability (r = .73) [33].

i. The Adult Resilience Scale (RS-14) consisting of 14 items recorded on a five-point Likert scale (0–4), with higher values indicating greater resilience tendencies in the general population [34]. This scale has good internal consistency ($\alpha = .85$) and good test-retest reliability ($r = .88$) [35]. This scale was used only for the young people aged 17 or older at service entry.

j. The Child and Youth Resilience Measure (CYRM-R) was obtained using 17 items recorded on a five-point Likert scale (1–5), with higher values indicating greater resilience [36]. This scale also has good internal consistency ($\alpha = .82$) and was used only for the young people aged 16 years or younger at service entry.

In summary, young people over 17 at service entry completed seven scales (AHS, ORS, CSE, EQ-5D, PHQ4, SIDAS, RS-14), and young people under 17 at service entry completed six scales (CHE, ORS, CSE, EQ-5D, PHQ-A, CYRM-R). In addition, in the discharge and follow-up surveys, all the young people completed purposely designed questions specific to the evaluation in order to better understand the service experiences of the young people. For example, the young people were asked to rate the helpfulness of the service and the service team on a 4-point Likert scale.

The discharge and follow-up online surveys also included a variety of open-ended questions that allowed the collection of qualitative data. These were questions about what was found to be most helpful about the service and suggestions for improvement. The same questions were used for the three follow-up one-on-one telephone interviews.

## Secondary data collection

Hospital service data were collected for HOPE service users for whom these data were available ($N = 70$). These data were used in the quantitative analysis, including part of a cost-benefit analysis for the new service. Hospital service data were collected for the three months before admission to the service, during admission, and for three and six months post admission, to establish to what extent hospital service use declined during and following service entry. These data considered a period of 3 months prior to service acceptance and up to 9 months after service acceptance, therefore covering the period from November 2021 to 30th September 2023, when all data collection ceased.

Changes in Crisis Assessment Treatment Team (CATT) calls, Emergency Department (ED) presentations, hospital admissions and nights in hospital were collected for the young people aged 17 or over ($N = 70$), and these were extrapolated over the 92 admissions to the HOPE service. It was not possible to obtain hospital service data for all these participants. Twenty were under the age of 17 and attending specialist children services, and two were referred from a different hospital network.

## Quantitative data analysis

**Tests for representativeness of primary data and attrition bias.** It was important to compare the characteristics of the evaluation sample ($N = 28$) with the characteristics of all the young people who entered the service before July 2023 ($N = 91$), in order to determine to what extent the primary evaluation sample was representative of the service population. The young people who participated in the evaluation were compared with those who did not in terms of age, gender and service use prior to service entry. This was done using chi-squared tests of association and appropriate mean comparisons (e.g., an independent samples t-test for age, a Negative Binomial Generalised Linear Model analysis allowing for over-dispersion for CATT call-outs).

Although 28 young people contributed primary data at some stage of the evaluation, only 24 young people provided primary data at entry, 14 young people provided primary data at discharge, and only eight provided primary follow-up data at 3 months and seven at 6 months. This made it important to test for any attrition bias, based on demographic and baseline measures as well as service use in the 3 months prior to entering the service. This was done using binary logistic regression analysis.

**Statistical analysis for primary data.** Descriptive statistics were derived separately for people under and over the age of 17 because different age-appropriately-validated scales were used for hope, resilience and psychological distress for these two cohorts. Linear Mixed Model analyses were then conducted to test for significant changes in these measures over time, with an assumption of normality for the residuals, supported by Kolmogorov-Smirnov and Shapiro-Wilk tests. A repeated measures ANOVA was not appropriate due to the incomplete nature of this primary data set. A random intercept was assumed for the larger over 17 sample ($N = 23$) and a fixed intercept for the very small under 17 sample ($N = 5$). In these models, age was controlled because age was found to be significantly associated with attrition at discharge. These Linear Mixed Model analyses assume that data is missing at random, while using maximum likelihood estimation to provide estimates for mean changes since service entry. Finally, descriptive statistics were presented, based on responses to questions related to the helpfulness of the service team and the intervention itself.

**Secondary data analysis for routinely collected hospital admissions.** The routinely collected hospital service use data described previously ($N = 70$), were used for the following purpose. A statistical analysis was conducted in order to determine whether there had been a decline in the use of hospital services by the young people during the 3 month intervention and in the 6 months following the intervention, compared to the 3 months prior to service entry. McNemar tests were used for this purpose.

## Cost-Benefit analysis using primary and secondary data

An indicative cost-benefits analysis was conducted using routinely collected hospital service data and publicly available proxy information (See Table A in S1 Text). As explained previously, the benefits in terms of reduced use of hospital services were calculated based upon the secondary data collected for the young people aged 17 or over ($N = 70$) and extrapolated over the 92 admissions which included 22 participants for whom it was not possible to obtain hospital service data. Quality of life improvement benefits were estimated based on the primary EQ-5D quality of life measure, completed only by the evaluation sample ($N = 28$), and the quality-adjusted life-year (QALY) scores suggested by Devlin et al. [37]. The results were then extrapolated across all 92 admissions and a sensitivity analysis was conducted based on the time for which benefits would be sustained.

A full costing approach was used to calculate the service costs. The costs included staffing, oncosts, operating, vehicle and accommodation costs. The detailed results of the cost-benefit analysis is provided in Table A in S1 Text, with the assumptions and data sources provided in Tables B to D of S1 Text.

## Qualitative data analysis

The open-ended questions in the discharge and follow-up online surveys and the three follow-up one-on-one telephone interviews were analysed using Braun and Clarke's 6-stage thematic analysis approach [38]. This type of analysis allows for a flexible inductive approach, grounded in the participant data. Initially, members of the research team became familiar with all responses before generating codes which were refined to create themes. The coding underwent

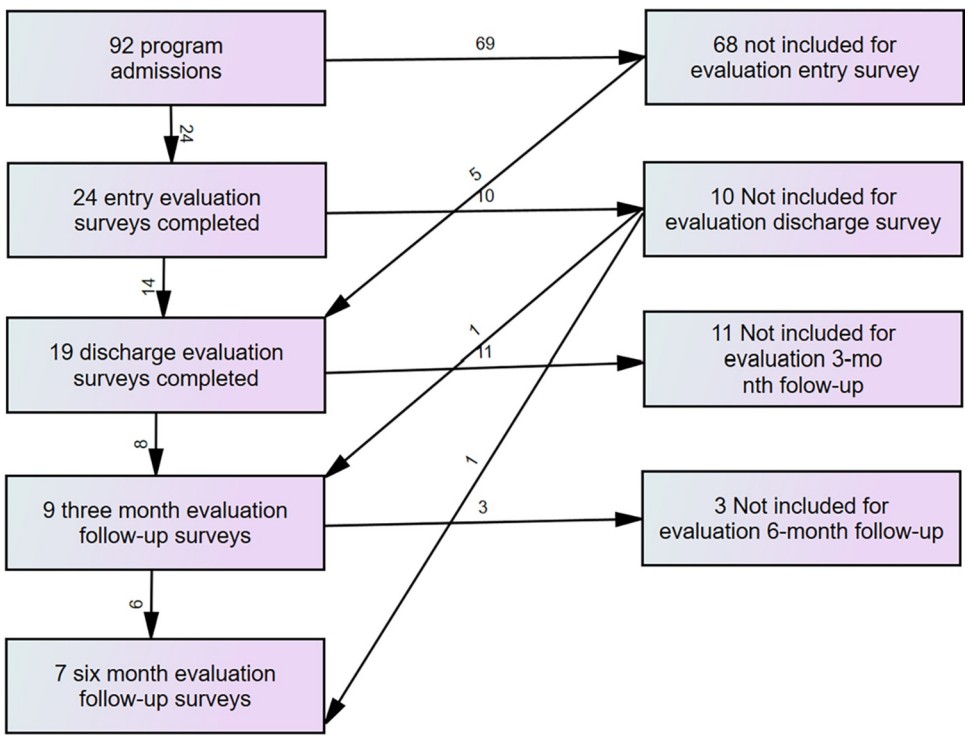

**Fig 1. Flow diagram for participation in the evaluation.**

independent checking by alternative members of the research team. Discrepancies were discussed until agreement was reached. Subsequently three researchers [initials removed for peer review] collated the themes and extracted key quotes to illustrate the themes.

## Results

The service started accepting referrals for young people from the middle of February 2022, with 107 referrals received before the end of June 2023. However, 16 young people were deemed ineligible for service for various reasons (for example, not in catchment), leaving 91. One young person was readmitted after completing the service so there were 92 admissions as shown in Fig 1.

The quantitative results of the evaluation are provided below together with the cost-benefit analysis results, followed by the qualitative results.

### Quantitative results

**Representativeness of evaluation sample.** The young people who were admitted to the service were referred by multiple sources including in-patient units ($n = 7$), emergency patient services ($n = 24$), CATT services ($n = 16$), other hospitals ($n = 10$), community organisations such as headspace and Stepping Stones ($n = 7$), schools (6) and family members (8). The age range at entry was 7–25 years old with a high proportion of LGBTQIA+ young people (48%) and culturally and linguistically diverse (CALD) young people (28%) as indicated in Table 1.

The evaluation sample matched the client service population reasonably well in terms of gender and age. Also, no significant differences were observed in terms of the mean number of CATT call-outs and emergency department presentations for those who did and did not participate in the evaluation. However, it seems that the CALD cohort was under-represented in

**Table 1. Demographic comparison of evaluation sample with overall service population.**

|  |  | Evaluation Sample N = 28 | Never in Evaluation Sample N = 63 | Service Population N = 91 | p-value |
|---|---|---|---|---|---|
| Age at entry | Mean | 19.7 | 19.9 | 19.9 | .806 |
|  | Standard Deviation | 3.1 | 4.3 | 4.0 |  |
|  | Range | 13–24 | 7–25 | 7–25 |  |
|  | Categories | Frequencies (%) |  |  |  |
| Age at entry | <17 years | 5 (18) | 15 (24) | 20 (22) | .527 |
|  | ≥17 years | 23 (82) | 48 (76) | 71 (78) |  |
| Gender | Males | 12 (43) | 26 (41) | 38 (42) | 1.00 |
|  | Females | 16 (57) | 35 (56) | 51 (56) |  |
|  | Non-binary | 0 (0) | 2 (3) | 2 (2) |  |
| Sexual Orientation | cis het | 11 (39) | 36 (57) | 47 (52) | .172 |
|  | LGBTQIA+ | 17 (61) | 27 (43) | 44 (48) |  |
| Ethnicity | ATSI+Maori | 4 (14) | 0 (0) | 4 (4) | .003# |
|  | CALD | 4 (14) | 21(33) | 25 (28) |  |
|  | Non-CALD | 20 (72) | 42 (67) | 62 (68) |  |
| Other | Neurodiverse | 5 (18) | 17 (27) | 22 (24) | .433 |

Cis het = heterosexual, ATSI = Aboriginal or Torres Strait Islander, CALD = culturally and linguistically diverse, # Cramer's V = .357.

the evaluation data (14% versus 28% overall), while the indigenous cohort was over-represented (14% versus 4% overall). The LGBTQIA+ cohort was also over-represented in the evaluation sample (61% versus 48% overall), but this difference was not significant.

**Overall impact in terms of routine hospital service utilisation data.** Considering only the young people over the age of 17 for whom routine hospital service data were available (N = 70), significant reductions in the numbers accessing hospital services were found after entry to the service, as indicated in Fig 2. In particular, McNemar tests showed that there were significant reductions in the number of young people requiring CATT call-outs during the

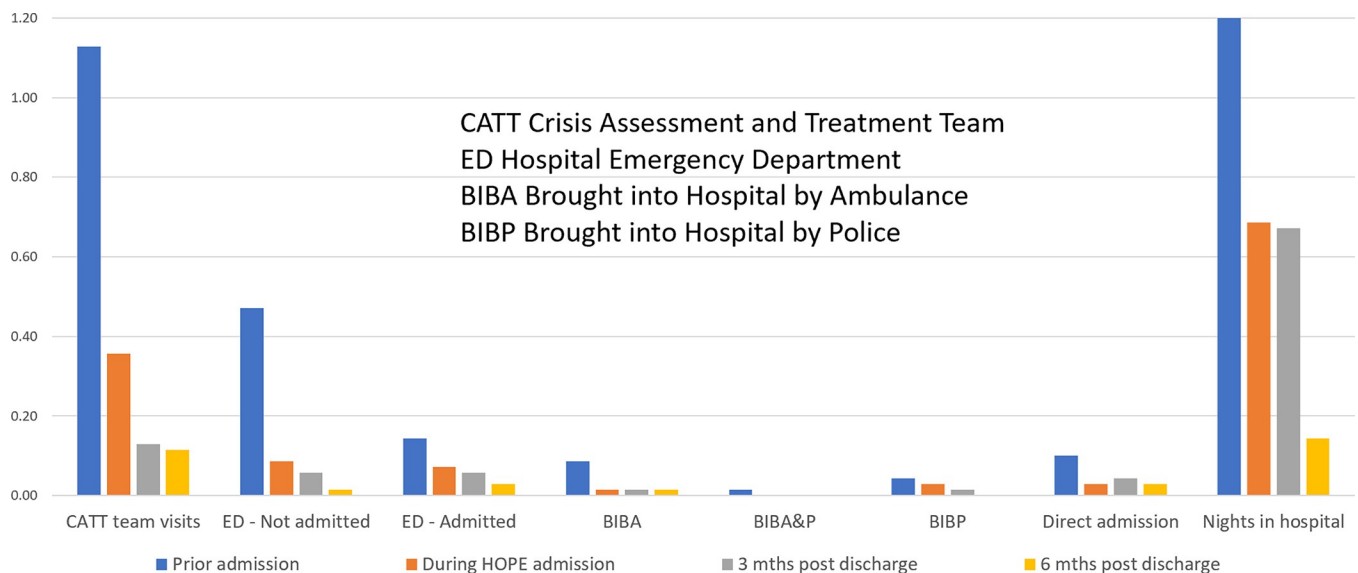

**Fig 2. Comparison of evaluation and non-evaluation samples in terms of hospital services required before, during and after entry to the service for young people over the age of 17.**

3-month service period, the following 3 month period and the three months after that ($p$ = .012, $p < .001$, p < .001 respectively). There were also significant reductions in the number of young people with emergency department admissions that did not lead to hospital admission ($p < .001$, $p < .001$, p < .001 respectively). There was less evidence in support of a reduction in emergency department admissions that led to hospital admission ($p$ = .031, $p$ = .070, $p$ = .004 respectively), with these being less common overall.

**Attrition from the evaluation study.** None of the baseline measures nor prior service use (CATT call-outs or ED admissions) were associated with attrition. However, attrition at discharge was significantly associated with age at service entry (Odds ratio = 2.046, $p$ = .020), suggesting that the odds of attrition at discharge more than doubled with each year of age at service entry.

**Statistics for evaluation of outcome scales for young people under 17 years old.** Outcome scales data were collected for only five young people under the age of 17 and these data were incomplete, with only 4 surveys completed at both entry and discharge from the service, 3 completed at 3-months follow-up and two completed at 6-months follow-up. The sample included 3 males and 2 females and the average age at entry was 15 years with a range of 14 to 16. Table 2 shows good scale reliability (Cronbach alpha>0.70) and the marginal means for each of these scales. No significant improvements were observed and the small sample suggests possibly unreliable results. However, there was a 16% increase in quality of life, a 13% increase in hopefulness, a 10% increase in resilience and an 11% increase in wellness (ORS) between entry and discharge.

**Statistics for evaluation of outcome scales for young people over 16 years old.** Data were collected for 23 young people aged 17 or older. Nine (9) of these young people identified as male and fourteen (14) identified as female. The age at entry ranged from 17 to 24 with a mean of 20.74 years and a standard deviation of 2.16 years. The results in Table 3 suggest significant improvements for several of the scales between entry and discharge; a 20% improvement in quality of life, a 17% improvement in PHQ4, a 15% improvement in resilience and a 31% improvement in SIDAS. However, the small numbers for 3- and 6-month follow-up meant that these results are likely to be unreliable.

**Descriptive statistics for helpfulness of service.** As shown in Tables 4 and 5, the service was generally seen as helpful, with the access given to additional services as the least helpful aspect. Only 5% of the young people who responded felt that their expectations were not met and did not feel confident to cope with problems, while 20% felt uncertain about still being burdened with suicidal thoughts. However, the majority of the respondents were very happy with the service received, and felt both more confident to cope with problems and less burdened by suicidal thoughts.

**Cost-Benefit analysis.** The cost-benefit analysis provided an indicative monetary calculation of the benefits the young people received from the service against the costs of the service.

**Table 2. Descriptive statistics for evaluation participants aged under 17 at service entry.**

|  | Cronbach Alpha | At Entry | At Discharge | 3 months Follow-Up | 6-months Follow-Up |
|---|---|---|---|---|---|
| Number of admissions |  | 4 | 4 | 3 | 2 |
| Outcome Rating Scale (ORS) | .874 | 264 (40.5) | 292 (40.5) | 266 (46.5) | 264 (60.1) |
| Poorer Quality of Life | .740 | 7.97 (.96) | 6.72 (0.96) | 6.33 (1.10) | 6.13 (1.42) |
| Health Questionnaire (PHQ_A) | NA | 19.36 (1.88) | 19.61 (1.88) | 18.67 (2.15) | 19.56 (2.78) |
| Coping Self-Efficacy | .977 | 141.6 (30.6) | 147.4 (30.6) | 133.7 (35.1) | 123.9 (45.4) |
| Child and Youth Resilience | .907 | 59.95 (6.68) | 66.20 (6.68) | 62.33 (7.66) | 56.71 (9.91) |
| Hope | .882 | 20.53 (2.91) | 23.28 (2.91) | 16.84 (4.10) | 18.05 (4.26) |

**Table 3. Descriptive statistics for evaluation participants aged 17 + at service entry.**

| | Cronbach Alpha | Marginal Means (Standard Errors) | | | |
| --- | --- | --- | --- | --- | --- |
| | | At Entry | At Discharge | At 3 months Follow-Up | At 6-months Follow-Up |
| Number of admissions | | 20 | 15 | 6 | 5 |
| Outcome Rating Scale | .909 | 238 (18.5) | 231 (20.6) | 219 (29.5) | 247 (32.3) |
| Poorer Quality of Life | .745 | 9.46 (.473) | 7.57*(.544) | 8.99 (.841) | 9.45 (.920) |
| Health Questionnaire (PHQ4) | .864 | 11.05 (.679) | 9.19#(.759) | 11.26 (1.098) | 8.84 (1.202) |
| Coping Self-Efficacy | .968 | 126.6 (10.12) | 136.8 (11.02) | 123.0 (14.89) | 134.6 (16.26) |
| Adult Hope | .914 | 26.89 (2.06) | 30.92 (2.27) | 23.59 (3.18) | 28.91 (3.48) |
| Resilience | .853 | 28.61 (2.06) | 32.85& (2.23) | 26.73 (2.93) | 32.73 (3.20) |
| Suicidal Ideation (SIDAS) | .732 | 28.00 (2.36) | 19.28^ (2.72) | 23.39 (4.22) | 25.40 (4.62) |

*Significant change since entry (p = .011)

#Significant change since entry (p = .036)

&Significant change since entry (p = .047)

^ Significant change since entry (p = .019)

It was found that, with an admission rate of only 5.6 young people in each month, the average service cost per admission exceeded the average benefit per admission by $751 over a 12-month period (See Table A in S1 Text). This rather low average intake rate can be ascribed to fluctuations in staff numbers and competing staff activities such as training. However, with the service now well established it may be possible to increase the admission rate, thereby improving the economic return of the service in the future. Limited studies on assertive outreach services, such as this, do suggest benefits can be maintained for at least 12 months, however, most have focused on suicide re-attempts as the outcome measure not cost-benefits [39]. If it can be assumed that these benefits can be maintained over two years, the average benefit per admission would have exceeded the average service cost by as much as $6,655 per admission.

## Qualitative results

Twenty-six young people provided qualitative feedback in relation to the service, verbally or in written form. This feedback was collected through a combination of interview and open-text survey data. In particular, there was feedback about what they found most helpful about the service and what could be improved. Results of the thematic analysis revealed 6 main themes around what young people found useful about the services. These were a) the benefit of outreach, b) feeling supported and understood, c) practical support, d) support for parents, e) less clinical approach, and f) service improvements.

**Table 4. Helpfulness of services offered by HOPE team as reported at discharge.**

| Service Offered | Total number responses | Frequency (%) | | | |
| --- | --- | --- | --- | --- | --- |
| | | Not at all helpful | Somewhat helpful | Moderately helpful | Very helpful |
| Initial Therapy Sessions | 20 | 0 | 2 (10.0) | 5(25.0) | 13(65.0) |
| Support worker | 16 | 0 | 0 | 2(12.5) | 14(87.5) |
| Family therapist | 6 | 0 | 0 | 1(16.7) | 5(83.3) |
| Access to Additional Services | 19 | 3(15.8) | 2(10.5) | 6(31.6) | 8(42.1) |
| | | Low (0–4) | Moderate (5) | High (8–9) | Very High (10) |
| Overall contact with HOPE Team | 20 | 0 | 1(5.0) | 7(35.0) | 12(60.0) |

**Table 5. Overall experience with HOPE team.**

| Experience | Total number responses | Frequency (%) | | | |
|---|---|---|---|---|---|
| | | Disagree | Uncertain | Agree | Strongly Agree |
| Easy to access help | 20 | 0 | 0 | 8(40) | 12(60) |
| Felt heard, understood and respected | 20 | 0 | 1(5) | 3(15) | 16(80) |
| Staff involved me in determining goals and strategies | 20 | 0 | 0 | 6(30) | 14(70) |
| Type of help as expected | 20 | 1(5) | 2(10) | 8(40) | 9(45) |
| Now more confident I can cope with problems | 20 | 1(5) | 2(10) | 6(30) | 11(55) |
| Feel less burdened with suicidal thoughts | 20 | 0 | 4(20) | 8(40) | 8(40) |

**Outreach.**   The theme of 'outreach' captured young people's appreciation for the outreach model of care offered in this service. Young people felt the outreach made contact more accessible by limiting issues associated with travel ("*I didn't have to travel to get this help*") and being met where they felt comfortable ("*we usually would meet up at a playground or a café*"), resulting in a less "*overwhelming experience*". For the young people, not going to a clinical environment made the experience less formal and more accessible, friendly and relaxed.

**Feeling supported and understood.**   Young people also really appreciated the emotional care and support they received from the workers in the service. They really valued the opportunity to just talk with someone about their issues, while feeling understood and supported. Having regular contact with the workers and being able to talk about their problems helped them feel less alone and better able to work through and process some of their feelings. The emotional support the young people felt took two distinct forms: a nonjudgemental space in which they could talk about their issues; and a sense of someone being there who could help them process their feelings.

An example of a comment illustrating the theme of being able to talk openly came from a young person who said:

"*You don't have a lot of adults you can talk to, so it was really good to be able to talk with people about it. Parents don't agree with a lot [of] shit, so having the team advocate was really good*".

Another noted that:

"*They let me be me. . . I felt like I was seen and heard*"

The sense that the service allowed young people to process their own feelings and work things out for themselves is illustrated by the young person who said:

"*just having someone to talk to and just kind of be figuring out how I was feeling and how I could actually help myself*".

For another young person, support came from:

"*Having someone to talk to each week who could support me and help me manage my stress and intrusive thoughts, just helping me process in a judge-free environment*".

**Practical support.**   Another common theme among the group was that they felt they received support in managing the day-to-day tasks and responsibilities that they had

previously struggled with. This could be divided into the subordinate themes of: *Managing day-to-day tasks* and; *organization and referrals*. Some young people felt that their mental health had previously got in the way of carrying out their day to day responsibilities and keeping their lives in order, so they found it very helpful to have support with managing these aspects of their lives. The practical support provided extended from things such as general organization and planning, to providing advocacy and referrals. One young person described how:

> "*It was helpful at the time with understanding and managing my day-to-day activities. They helped me with a lot of information and planning. . .like printing stuff out and helping me with applications, or with organising meetings with my parent. The. . . service actually helped apply stuff physically and helped me put changes through in my life, instead of just seeing them once a week, and that's it*".

Another discussed how a member of the team had helped her with practical social and financial issues:

> "*She was a big saving grace,. . . saving me from losing my house*".

For other young people, dealing with the complexity of the mental health system was made easier by the support of the new service:

> "*I really struggled to organise and reach out for different help services and therapy sessions and what not, looking at different options for my mental health, and they were able to guide me through that and help find someone that I'd like. The regular visits helped with organisation of appointments and other services*".

**Support for parents.** Beyond the direct support the young people experienced, three young people (under 17 years) identified the importance of the service providing support to their parents:

> "*The support towards the parents. I feel like with these mental health issues, um, a lot of it comes from the environment and parents play a large role in that, so, having the team there to coach the parents through and give them their own mini therapy*".

By providing support to the parents, young people felt that they were getting vicarious support by addressing the well-being of the family system.

> "*I think that its role in helping my parents did a lot to help me*".

**Less clinical approach.** In general, something the young people appreciated was that the care they received felt less clinical compared to what they had previously experienced. They felt that having the service team members see them in environments outside of the clinic, and engaging with them in a more egalitarian way, was a strength of the service which allowed young people to feel more comfortable:

(I felt) *"really calm and it didn't feel like there was too much pressure to, you know, talk"*.

(The service was) *"A LOT less daunting than going to a psych ward, and much more approachable for those who are struggling with mental health and scared about the extreme clinical work, like psych ward and CATTs (Crisis and Assessment Teams)".*

*"I think probably the fact that I wasn't going into an office and it was, it wasn't incredibly formal. . . just relaxed and calm. It was really good".*

Having service team members with their own lived experience of mental health issues also helped to reduce the power imbalance and made the young people feel like they were talking to someone who directly empathized with them, providing a safe supportive space. The lived experience of some of the service team members also served as a helpful reminder that people can get better:

*"that there are people that have come out the other side healthy, and haven't died".*

The lived experience worker enabled the development of a level of trust between the treating team and the young person:

*"having someone on the team that has been through something like that before is very helpful, and the level of trust they had with me. I felt like I could talk freely as it wasn't going to be like —'you still have suicidal ideation, you need to go to hospital'. It was more like 'are you going to act on them?', no, well do you need someone with you to make sure you don't?'. This was as opposed to like, 'you're feeling that way, you're just going to have to go into 24hour care kind of thing'".*

This reduction of the power imbalance between client and staff enabled the trust and rapport to be firmly established.

**Service improvements.** Twelve of the young people felt there was nothing that needed to be improved about the service, with two young people describing the service as "*perfect*" because "t*hey gave me everything I needed"*. The main suggested area for improvement related to the transition from the service to another service at the end of the 3-month period of care. This aspect was mentioned by five young people who were all over the age of 17 years. These young people felt that there was room for improvement in the context of moving from receiving structured support to the "*transition*" or "*re-adjusting to living a normal life*". In particular, these young people described this as being a challenge because, as one of them mentioned, *"I am not good at organising stuff"*.

Other improvements suggested involved the frequency of sessions and duration of the service. Some felt having more and longer sessions would have been beneficial:

*"I just felt that they (the sessions) were quite short and quick"*

*"I felt that there was, you know, more that needed to be talked about".*

Also, a longer period of care stretching beyond 3 months was suggested by some of the young people, who

*"felt that it was a little too short. . . I'm still searching for help".*

While these last suggestions are common to time-limited support services, the overall qualitative data indicates that the model of care which was chosen through a co-design process for

the implementation of this new service is successful in meeting the needs of the cohort of young people who are presenting for care. In particular, the move away from traditional, medically or psychologically centred care, offered in a clinical setting, was greatly appreciated, allowing the young person to feel supported and held in their usual environment. Support from workers with lived experience for both the young person and their parent(s), as well as other forms of psychosocial and community-based support was welcomed, as it helped the young person and family feel that they were able to get on with their lives. While the service also provides clinical support through traditional models such as medication reviews and access to psychological therapies, the young people were less likely to mention these aspects of the service as ones which stood out as being especially helpful. This may be because they more closely conform to what the young person may have expected to receive, and were thus less likely to arouse comment.

## Discussion

The objective of this evaluation was to report on the experience of the young people who entered the service in its first 17 months, particularly considering the effects on suicidal ideation, coping self-efficacy, resilience and well-being. Overall, the study has indicated significant improvements in suicidal ideation, resilience and well-being with significant sustained reductions in the number of CATT team call-outs and ED presentations in the 9 months following entry to the service. In addition, significant improvements in quality of life have been observed and a cost-benefit analysis has suggested that the service may be economically justified if these benefits can be retained over more than a 12 month period.

### Quantitative findings

The evaluation sample ($N = 28$) used for much of the evaluation was small but reasonably representative of the overall sample of young people entering the service up until the end of June 2023 ($N = 91$). However, recruitment was difficult with only 24 of the evaluation sample able to be recruited at entry. A Finnish study of health survey response rates shows that response rates do tend to be lower for younger than older people and for people who are unwell, so this should not have been a surprise [40]. Attrition was also high with only 19 of the 28 young people able to be contacted at discharge, and 9 and 7 young people at the 3-month and 6-month follow-ups respectively. The more than doubling of the attrition rate for each year of age suggests that there may be some age-related bias in the results (perhaps employment-related bias), and for this reason age was controlled for in the linear mixed model analyses. Lower response rates related to age have also been observed in an English national inpatient survey with a response rate of 64% in an adolescent ward and 52.3% for young people in an adult ward [41].

Overall, the study indicated significant reductions in suicidal ideation and psychological distress at discharge. The high rate of attrition makes it impossible to determine whether these improvements were sustained 3 and 6 months after discharge. Although the reduction in CATT call-outs and ED presentations in the months following discharge suggest that this is likely, it is important that more data is collected post discharge in the future.

Improvements in terms of increased resilience are also most encouraging. Increased resilience is strongly related to reduced suicidal ideation in young people, making these changes especially promising [42]. Resilience building has been particularly recommended by the Crisis and Trauma Resource Institute (CTRI) for young clients who are suicidal [43].

The improvements in quality of life are incorporated in the cost-benefit analysis conducted for the service, together with the implications for CATT call-outs and ED presentations. However, our cost-benefit analysis does not take into account the likely reduction in suicide that is

achieved by the service, suggesting that we have under-estimated the service benefit. The total economic cost of suicide by young people aged 15–24 in Australia has been estimated at $511.1m per year based on research conducted in 2014 [44].

However, the quantitative results also point to other areas that could perhaps be improved in the future. The scores for hopefulness, the Outcome Rating Scale (ORS) and coping self-efficacy showed no significant improvement. Hopefulness has been linked to reductions in suicidal ideation, making improvements in hopefulness an important future priority for the service, ideally also leading to improvements in the ORS [45]. However, the ORS taps into the areas of personal well-being, interpersonal interactions and social relationships, perhaps suggesting that more work is required in these areas. Finally, coping self-efficacy considers the ability of the young person to manage their emotions, their mental health, their relationships and their impulses. Associations have been found between health-related self-efficacy and suicidality, making this another protective factor that can reduce suicidal ideation, particularly in young males [46, 47]. The lack of a significant improvement in these measures potentially indicates that a longer or more focused service of care is needed in order to improve the chances of achieving long-term recovery.

## Qualitative findings

The qualitative findings align with the Crisis and Trauma Resource Institute (CTRI, 2023) resilience building framework, helping to provide an explanation for the success of the service in reducing suicidal ideation [43]. The relaxed nature of an outreach service and the emotional support provided by the service team clearly resonated with the young people, speaking to one of the methods often used to build resilience in suicidal young people; namely "normalize their thoughts and feelings". In addition, advice received in this non-judgemental space helped them to better understand and manage their emotions, speaking to a second resilience building method (CTRI, 2023); "help them learn to ride the wave of emotion" [43]. A third resilience building capacity, "focus on helping them learn new skills" was also found to be helpful. In particular, skills for managing day-to-day tasks and responsibilities, and help in finding appropriate mental health services ensured that real changes could be made. A fourth resilience building capacity, "include their support system", was particularly addressed through the support provided to parents and carers. However, one of the resilience building methods often recommended (namely "explore their protective factors and reasons for living") was not clearly enunciated by the young people.

Twelve of the young people felt there was nothing that needed to be improved about the service, however, extending the service and better managing the transition to external services were seen as important deficiencies by several of the young people. It is worth mentioning that the co-operation of the young people was required in preparing them for discharge from the service. Some of the young people may have been too busy, perhaps too overwhelmed, to take up all the opportunities that were offered.

However, it is clear that a new kind of service has been provided which is seen as helpful by the young people included in this study. The priorities that emerged from the co-design process in regard to the importance of peer support for the young person, their support network as well as psycho-social support, have clearly resonated with these young people. This suggests an effective implementation of these aspects of the design despite the delays, staffing changes and conflicting stakeholder expectations that may have acted as implementation barriers [19]. However, although the codesign process did recognise the importance of engagement with external services as soon as possible, this aspect of the design appears to have been less well implemented, suggesting an area where more work is required.

This evaluation is more comprehensive than evaluations of similar co-designed services found in the literature, including an economic evaluation as well as both quantitative and qualitative analyses [18]. The service itself is also perhaps more comprehensive, incorporating peer support for family as well as the young people themselves, and providing psycho-social support as well as clinical support.

## Implications

As indicated by the qualitative data collected, the overall success of the service can certainly be partly ascribed to the co-design outcomes which set the stage for a service that emphasised outreach, peer support for the young person and family and the need for continuity of service after discharge from the program. This speaks to the importance of co-design for interventions of this nature.

The findings clearly support the importance of non-clinical as well as clinical support for these young people and their families. Traditional pathways of emergency department or inpatient psychiatric treatment need to be followed by multi-disciplinary, collaborative, peer-inclusive outreach, making the development of a workforce for this purpose an important future priority. In addition, an integrated and coordinated service system, which catches young people at critical junctures in their mental healthcare journey, such as at discharge from an inpatient unit, will be better placed to stop young people falling through the cracks. Having a service which builds-in referral and handover to ongoing care, helps to develop a better integrated system across hospitals, community care, schools and other services.

The evaluation team experienced difficulties in recruiting young people for this study. Traditional research methods for contacting the young people were often not successful, suggesting that alternative approaches, perhaps via social media platforms that young people frequent, would have resulted in more participation from young people. Closer adherence to the ideals of co-evaluation, with the inclusion of a young person with lived experience of suicide on the evaluation team, is therefore recommended for future studies.

Finally, in view of the current interest in eHealth delivery of services of this nature it is important to consider what, if anything, this evaluation has to say about how such eHealth services are best implemented. Ideally, such services must allow for outreach on a personal level, ensuring that post-suicide support is provided in a relaxed environment, incorporating emotional support by peer and family support workers, alongside therapeutic supports. The challenge will be creating an eHealth environment that is just as good as meeting in a park or on a basketball court, or whatever works for the young person.

## Limitations

This evaluation has several limitations including the small sample size, particularly the very small number of young people who completed the follow-up surveys at 3 and 6 months. The age bias shown in the attrition rate may be due to employment pressures, also perhaps pointing to an over-burdensome survey questionnaire, but this needs further investigation. It is also possible that the young people prefer to move directly back to the life they had previously, without wanting to reflect on their recovery journey. We have no way to determine what the follow-up values would have been for those who did not respond at follow-up. There was also bias in the evaluation sample in terms of CALD and indigenous representation which may have affected the results, and the small numbers do not allow further investigation of this effect.

Our cost-benefit analysis also has limitations. Routinely collected hospital service data could not be collected for 22 of the 91 young people who entered the service and it was

assumed that these costs were the same as for the other admissions. Secondly, the calculation of benefits in this study relied upon hospital service data for the three months prior to admission being representative of the potential savings over a year, and this may be over-stated. However, some benefits may also have been under-stated because any ambulance attendances at home and any ED attendances accompanied by friends or family were excluded. In addition, the study did not incorporate the costs of suicide or any improvements in quality of life for the young people's primary carers, which may have resulted in additional benefits.

Our qualitative analysis also has limitations in that we partially relied on online surveys for the collection of these data. Although online surveys are acceptable for the collection of qualitative data, this tends to result in less rich data. It also means that there is no opportunity to ask follow-up questions or prompt someone to elaborate.

The small sample size problem is likely to exist for any evaluation study involving young people with serious mental health problems, and for this reason and future service development it is recommended that such evaluations be continued over a longer period of time. In this way, it should be possible to obtain more accurate data and a better reflection of what the service can achieve in economic terms in the longer term.

Finally, the results of this evaluation are limited to the service as presented during 2022 and 2023. This followed the very severe COVID-19 lockdowns that took place in Melbourne during 2021 which are known to have severely impacted the mental health of young people [48]. Also, the design and implementation characteristics of the service were clearly very specific and may not generalise to other services of this nature.

## Conclusions

It is hoped that the findings of this evaluation will be helpful for those wanting to implement other services of this nature in other locations. In particular, it is suggested that the principles of co-design, peer support and planning for future service provision be incorporated in any such service, regardless of whether the service is offered using eHealth or face-to-face. This evaluation has shown that a service of this nature can do much to improve the lives of young people at risk of suicide but, as indicated by the above study limitations, there is certainly room for improvement. Further co-evaluation research in this area is therefore warranted, to ensure that the voice of these young people is better heard and understood.

## Supporting information

**S1 Text. Cost-Benefit analysis.** Table A in S1 Text: Cost-benefit analysis at 9 and 12 months. Table B in S1 Text: General assumptions. Table C in S1 Text: Cost item data sources and assumptions. Table D in S1 Text: Benefit values–data sources and assumptions. (DOCX)

## Acknowledgments

The authors would like to thank and acknowledge all those who participated in the co-design process and research component of this study, in particular the young people with a lived experience who gave up their valuable time and expertise. We acknowledge that this work was carried out on the lands of the people of the Kulin Nation and we pay our respects to their culture and their elders past, present and emerging.

## Author Contributions

**Conceptualization:** Denny Meyer, Liza Hopkins, Michelle Kehoe, Maja Nedeljkovic.

**Data curation:** Liza Hopkins, Michelle Kehoe, Richard Whitehead.

**Formal analysis:** Denny Meyer, Liza Hopkins, Michelle Kehoe, Richard Whitehead, Kathleen de Boer, Debra Osborne, Maja Nedeljkovic.

**Funding acquisition:** Denny Meyer, Liza Hopkins, Michelle Kehoe, Maja Nedeljkovic.

**Investigation:** Denny Meyer, Liza Hopkins, Michelle Kehoe, Richard Whitehead, Debra Osborne.

**Methodology:** Denny Meyer, Liza Hopkins, Michelle Kehoe, Richard Whitehead, Kathleen de Boer, Debra Osborne, Maja Nedeljkovic.

**Project administration:** Denny Meyer, Liza Hopkins.

**Resources:** Denny Meyer, Liza Hopkins, Michelle Kehoe, Richard Whitehead.

**Software:** Denny Meyer, Liza Hopkins, Michelle Kehoe, Richard Whitehead.

**Supervision:** Denny Meyer, Liza Hopkins.

**Validation:** Denny Meyer, Liza Hopkins.

**Visualization:** Denny Meyer.

**Writing – original draft:** Denny Meyer, Liza Hopkins, Michelle Kehoe, Debra Osborne.

**Writing – review & editing:** Denny Meyer, Liza Hopkins, Michelle Kehoe, Richard Whitehead, Kathleen de Boer, Debra Osborne, Maja Nedeljkovic.

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
