## [Decision Letter · Decision Letter 0]

27 Feb 2024

PMEN-D-24-00045

Evaluation of a service introduced to support young people at risk of suicide using a cohort design, a mixed methods analysis and cost-benefit analysis

PLOS Mental Health

Dear Dr. Meyer,

Thank you for submitting your manuscript to PLOS Mental Health. After careful consideration, we feel that it has merit but does not fully meet PLOS Mental Health’s publication criteria as it currently stands. Therefore, we invite you to submit a revised version of the manuscript that addresses the points raised during the review process.

We look forward to receiving your revised manuscript.

Kind regards,

Justus Uchenna Onu, MBBS, FWACP, FMCPsych

Academic Editor

PLOS Mental Health

Journal Requirements:

1. Please provide separate figure files in .tif or .eps format only and remove any figures embedded in your manuscript file. Please also ensure all files are under our size limit of 10MB.

https://journals.plos.org/mentalhealth/s/figures 

https://journals.plos.org/mentalhealth/s/figures#loc-file-requirement

2. We have noticed that you have uploaded Supporting Information files, but you have not included a list of legends. Please add a full list of legends for your Supporting Information files after the references list.

Additional Editor Comments (if provided):

Thank you for submitting your manuscript.

As clearly pointed out by the reviewer, the authors need to ensure the clarity of the paper.

In addition to the comments raised by the reviewer, the following comments from the editor should be considered:

1. Clearly specify the research questions/objectives intended to answered in the study

2. Clearly specify the methodological process(es) to answer each of them. This includes, the objectives to be addressed using the quantitative and qualitative designs, the data analytic method method used for each the objective or research question. Why was mixed linear model used instead of Repeated measures ANOVA considering the four points of measurement for same sample.

3. The presentation of the result should reflect the questions raised or the objectives to be addressed.

4. The authors should state the reasons given for refusal to participate and the huge loss during the follow-up possibly using a flow diagram for clarity

5. The penultimate and last assessment was only done in about 4-5 persons from the initial 21. Do the authors think that this number is enough to answer the questions of improvement in quality of life, well-being and reduction in suicidality score? While not consider using the Last Observation Carried Forward Option. Clearly justify that the number lost to follow-up will not affect the overall findings.

4. Provide reference for many assertions such as the first sentence in the introduction, "Suicide is now the leading ................................. in Australia".

Reviewers' comments:

Reviewer's Responses to Questions

**Comments to the Author**

1. Does this manuscript meet PLOS Mental Health’s publication criteria? Is the manuscript technically sound, and do the data support the conclusions? The manuscript must describe methodologically and ethically rigorous research with conclusions that are appropriately drawn based on the data presented.

Reviewer #1: Partly

2. Has the statistical analysis been performed appropriately and rigorously?

Reviewer #1: I don't know

3. Have the authors made all data underlying the findings in their manuscript fully available (please refer to the Data Availability Statement at the start of the manuscript PDF file)?

Reviewer #1: No

4. Is the manuscript presented in an intelligible fashion and written in standard English?

Reviewer #1: Yes

5. Review Comments to the Author

Reviewer #1: Thank you for the opportunity to review the manuscript titled: 'Evaluation of a service introduced to support young people at risk of suicide using a cohort design, a mixed methods analysis and cost-benefit analysis'. The paper reports results of a worthy area of research and evaluation, that of preventing suicides in young people. I have made the following comments which I hope are helpful.

Abstract:

The abstract would benefit from being clear regarding the dates of study and sample sizes obtained for the qualitative and quantitative data collection elements. It is also unclear whether the quantitative element is secondary data analysis or primary data collection; please can this be made more explicit.

Introduction:

Para 1: context is provided by %s of young people dying by suicide, but what is the n associated? This will help to provide a clearer context.

Thank you for clear context and rationale for the study on Pages 5-6, this makes it clear for the reader.

Page 6 lines 108-110 can you reiterate which service, where that is based, and reach of the service. Also, it states here that this is an evaluation. Immediately following it states it's a co-designed service. Does the evaluation evaluate the co-design process, the service, or both? Please can this be made clearer. If it does not include an evaluation of the co-design process then there appears to be a lot of detail provided on the co-design aspect and challenges and not a lot of detail and context on the actual service, which is probably more useful and should be inserted if the evaluation focused on the service itself.

It is unclear what the outcome measures were for the evaluation; clarity on these would be needed in order to understand the results of the evaluation.

Methods:

It is stated "However, the recruitment of 150 young people for this evaluation study began on 5th April 2022" - can you make it clearer what the time period for the data analysed was. It implies no data before April, but can this be made clearer.

I assume, but this could be made clearer, that all young people who came through the service were approached to take part in an interview? Or was there a sampling frame? If yes, what? Please make this clearer.

Also see comment in abstract about quantitative methods being unclear. This is the same on Page 8 lines 164-175. Please can the methods be made much clearer.

Page 9 details Hospital Service Measures and others. Lines 178-183 in particular implies that the data collection period was different to the period April to Sept as stated earlier. Please can all data collection measures, methods, and data time points be made much clearer. This section also suggests a component of secondary data analysis and ties in with other comments around a lack of clarity on the methods used and data collection approaches.

Re the measures, a lot of scales were used. Was there any feedback or thought given to participant burden? How long did it take for these to be completed by participants?

Analysis:

Page 13, lines 289-303 is not about analysis. This section needs moving to earlier in the methods as it's more about the design of the data collection instrument and not the analysis. It does also remind me that the methods are very sparse and means it would be infeasible for someone to fully know what was done, when, how and why. I would recommend revisiting Methods in terms of sampling, recruitment and data collection procedures and to make it clear for all components of the research.

Results:

This "Of these young people sixteen were declined service for various reasons" does not make sense and needs re-wording.

"One young people" should be 'One young person'.

Page 15 Line 323 all acronyms here and elsewhere need defining in full first time stated.

Page 20 states "There were several open-ended questions in the discharge and follow-up surveys that allowed the voices of the young people to resonate with their closed question responses." No where earlier in the methods information does it say qualitative data included open responses from a survey. Please ensure methods, including analysis, are clear throughout.

Are the interview themes a combination of interview and open-text survey data or just interview data? Please make clear.

It is somewhat confusing having some interview quotations in tables and some in text. Why is this? Can it be made consistent? There is also very limited narrative about themes and sub-themes. I appreciate word counts of the journal but it almost comes across as qualitative findings/narrative is somewhat limited/rudimentary.

It is hard to contextualise the qualitative responses/findings because there is very limited context provided earlier in the paper on the service, it's components etc.

Discussion:

I think this sentence is somewhat far-reaching given the results earlier presented "In addition, significant improvements in quality of life have been observed and a cost-benefit analysis has indicated that the service is also likely to be economically justified if these benefits can be retained over more than a 12 month period.".

Is there any evaluation data that could suggest reasons for attrition? Presumably this is also linked to their current mental condition and mental capacity to participate? Any data/findings to comment on here?

It is hard to contextualise the discussion because there is very limited context provided earlier in the paper on the service, it's components etc. It is difficult to see which service elements are well thought of or not and how widespread or not both positive and negative views were.

Implications:

Implications seem to centre on co-design challenges, yet no results are actually presented which touch on co-design. Please ensure implications closely follow findings reported in this paper and not elsewhere.

6. PLOS authors have the option to publish the peer review history of their article (what does this mean?). If published, this will include your full peer review and any attached files.

**Do you want your identity to be public for this peer review?** For information about this choice, including consent withdrawal, please see our Privacy Policy.

Reviewer #1: No

---

## [Editor Report · Decision Letter 1]

6 May 2024

Evaluation of a service introduced to support young people at risk of suicide using a cohort design, a mixed methods analysis and cost-benefit analysis

PMEN-D-24-00045R1

Dear Professor Meyer,

We are pleased to inform you that your manuscript 'Evaluation of a service introduced to support young people at risk of suicide using a cohort design, a mixed methods analysis and cost-benefit analysis' has been provisionally accepted for publication in PLOS Mental Health.

Best regards,

Justus Uchenna Onu, MBBS, FWACP, FMCPsych

Academic Editor

PLOS Mental Health

- In your ethics statement, please ensure that you name the precise hospital and university human research ethics committees that gave approval.